# Alternative Tissue Sampling for Improved Detection of *Candidatus* Liberibacter asiaticus

**DOI:** 10.3390/plants12193364

**Published:** 2023-09-23

**Authors:** Subhas Hajeri, Sandra Olkowski, Lucita Kumagai, Neil McRoberts, Raymond K. Yokomi

**Affiliations:** 1Citrus Pest Detection Program, Alliance of Pest Control Districts, Tulare, CA 93274, USA; 2Department of Plant Pathology, University of California, Davis, CA 95616, USA; nmcroberts@ucdavis.edu; 3California Department of Food & Agriculture, Sacramento, CA 95832, USA; lucita.kumagai@cdfa.ca.gov; 4Agricultural Research Service, USDA, SJVASC, Parlier, CA 93648, USA; ray.yokomi@usda.gov

**Keywords:** HLB, citrus greening, qPCR, early detection, detection sensitivity, sample tissue, leaf petiole, fruit peduncle, feeder root

## Abstract

Early detection and prompt response are key factors in the eradication of ‘huanglongbing’ (HLB) in California. Currently, qPCR testing of leaf tissue guides the removal of infected trees. However, because of the uneven distribution of ‘*Candidatus* Liberibacter asiaticus’ (*C*Las) in an infected tree and asymptomatic infection, selecting the best leaves to sample, from a mature tree with more than 200,000 estimated leaves, is a major hurdle for timely detection. The goal of this study was to address this issue by testing alternative tissues that might improve the *C*Las detection rate. Using two years of field data, old and young leaves, peduncle bark of fruit, and feeder roots were evaluated for the presence of *C*Las. Quadrant-peduncle (Q-P) tissue sampling consistently resulted in better *C*Las detection than any other tissue type. Q-P samples had a 30% higher qPCR positivity rate than quadrant-leaf (Q-L) samples. No significant seasonal patterns were observed. Roots and single peduncles had similar detection rates; both were higher than single leaves or Q-L samples. If symptoms were used to guide sampling, 30% of infected trees would have been missed. Taken together, these results suggest that Q-P tissue sampling is the optimal choice for improved *C*Las detection under California growing conditions.

## 1. Introduction

California is the top citrus-producing state in the USA, with production worth over USD 3 billion and over USD 7 billion in total economic impact on California’s economy [1] ‘Huanglongbing’ (HLB) is the most devastating disease of citrus worldwide, with no cure, and all commercial citrus cultivars are susceptible to the disease, particularly sweet orange and grapefruit [2,3]. The phloem-limited bacterium ‘*Candidatus* Liberibacter asiaticus’ (*C*Las) is associated with HLB, and transmitted by the Asian citrus psyllid (*Diaphorina citri* Kuwayama).

In the USA, HLB is well established in major citrus-growing states, such as Florida, Texas, and California. In Florida, where HLB was first detected in 2005, the disease has spread throughout the state, costing more than USD 7.8 billion in revenue, 162,200 citrus acres, and 7513 jobs since 2007 [4]. It is estimated that even a 20% reduction in California citrus acreage would cause a loss of 8213 jobs, USD 214 million in employee income, and reduce state GDP by USD 569 million [1]. In California, 6009 *C*Las-positive trees have been removed to date, but no HLB-positive trees have been detected in any commercial citrus orchard. HLB is currently limited to residential properties in five southern California counties: Los Angeles, Orange, Riverside, San Bernardino, and San Diego [5].

Leaf sampling is a relatively inexpensive way to conduct large-scale surveys to estimate plant disease incidence [6,7]. Mature leaf sampling is the regulatory standard sampling method adopted in California, based on Florida experience [8,9]. Quantitative PCR is the current standard regulatory method for *C*Las detection. However, collecting the right leaves to test from a mature infected tree is a major challenge due to the uneven spread of the pathogen and variation in symptom development. Some infected trees may be asymptomatic for months to years after infection or the symptoms may be barely visible in the dense canopy. Additionally, some trees exhibit symptoms that can mimic other biotic or abiotic symptoms. Florida’s citrus industry relied on scouting for symptomatic leaves, followed by qPCR testing to detect HLB. Currently, 90% of the citrus groves in Florida are infected [10]. One of the main reasons for the current state of the Florida industry is believed to be the delayed response in tracking the ACP/HLB outbreaks. The California citrus industry is continuing its efforts to validate methods for early detection of *C*Las, which is critical in mitigating the spread of the disease.

‘Early detection’ refers to the detection of infected trees before any visible symptoms appear. However, owing to the current non-availability of an early detection tool (either direct or indirect), the best option available is to optimize tissue sampling. The validity of qPCR tests using RNR probes has already been demonstrated [11], but, to obtain consistent results, it is important to determine the most reliable tissue for *C*Las testing.

The goal of this study was to identify which plant tissue types are most effective for detecting *C*Las. Effectiveness can also be expressed as the detection sensitivity of the tissue type, which is the proportion of qPCR-positive samples out of the total known positives, or, in other words, the likelihood of detecting the pathogen in a tree that is known to be infected. In this study, data from field-collected samples over the course of two years were used to evaluate the relative detection sensitivity of the petiole of old and young leaves, the peduncle bark of fruit, and feeder roots. The results indicate that quadrant sampling of peduncle tissue significantly increases the probability of detecting *C*Las in an infected tree, compared with sampling other tissues, including quadrant-sampled leaves.

## 2. Results

### 2.1. CLas Detections by Tissue Type

Of the 765 sampling events conducted during the study, 48% had at least one qPCR-positive result from the tissues collected (Table 1). Root and single-peduncle samples had a marginally higher count (Figure 1A) and percentage (Table 1) of positive results than old or young leaves. Overall, the detection rates of root and single-peduncle tissue results were similar, with 35% and 34% positivity, respectively. For the time period during which quadrant-peduncle samples were collected (April–October 2021), this tissue type had the highest count (Figure 1B) and the highest percentage of *C*Las-positive detections. Old leaves had the lowest percentage of detections (30%) out of the study tissue types.

### 2.2. Detection Sensitivity of Tissue Types and of Symptoms

Detection sensitivity (Se), i.e., the likelihood of detecting the pathogen in an infected tree, was highest for quadrant-peduncle (Q-P) samples. The overall Q-P Se was 84% (172/206) (Table 2) during the 7.5 months in year 2 that these samples were collected. The Q-P samples also had the highest Se among the three seasons tested (see Section 4.3.2. for details on seasonal effects results). Among tissue types that were collected throughout the study, single-peduncle tissue and root tissue had the highest Se (75% and 76%, respectively). All leaf tissues—old and young leaves, single and quadrant samples—had a lower Se than peduncle and root tissues, in every season (Table 2).

The presence or absence of symptoms was recorded for 601 trees. This was compared with qPCR-positive/-negative results from all plant tissues considered together, i.e., if any of the regulatory and/or study tissue samples collected during the same sampling event tested positive, the tree was coded as positive (*n* = 260). The presence of symptoms had a 70% (182/260) detection sensitivity (Table 3). Thus, 30% (78/260) of infected trees would not have been qPCR tested if symptoms had been used to guide sampling.

Quadrant-leaf (Q-L) samples for regulatory use were collected by CDFA field staff at 84% (642/765) of the sampling events during the study period (Table 1). Of these, 25% (162/642) were *C*Las+. The overall detection sensitivity of these samples was 72% (162/224) for known-positive trees with a *C*Las-positive result from a previous time point. Similarly, when comparing Q-L results with study tissue samples collected simultaneously and under the same conditions, Q-L samples had a 30% (69/231) false-negative rate in trees that were otherwise identified as being infected (Table 4).

In cases where all four Q-L samples tested negative (*n =* 479), roots had the highest positivity rate out of any of the sample tissues, i.e., 78% (Figure 2A). When one or two Q-L samples tested positive (one positive: *n =* 36; two positive: *n =* 36), peduncle samples had the highest positive percentage (87% and 97%, respectively) (Figure 2B,C). When three or four Q-L samples tested positive (three positive: *n =* 28; four positive: *n =* 63), all single sample tissue types showed 100% positive detections apart from roots, which dropped to 93% and 81%, respectively (Figure 2D,E).

#### 2.2.1. Seasonal Effects

The majority of *C*Las-positive trees was detected in the summer and fall seasons (Figure 3A). Peduncle and root tissue had consistently higher qPCR-positive results than old or young leaf tissue, in all seasons. Single-peduncle samples and root-tissue samples collected across the full time series were observed to have similar positive counts. Although they differed somewhat between seasons, neither had consistently higher counts. The results showing detection sensitivity likewise demonstrated that single-peduncle tissue and root tissue were similar overall (74.6% vs. 76.3%, respectively), but with small variations between them across seasons (Table 2 and Figure 4).

There were no statistically significant associations between tissue type and season, when these interactive effects were statistically modeled (see Section 4.3.3. for more modeling results).

Quadrant-peduncle samples (*n =* 314) were collected in the spring, summer, and fall seasons of year 2. During this time period, Q-P samples had consistently higher *C*Las-positive counts than any other tissue sample type (Figure 3B). They also had a consistently higher likelihood of detecting infected trees both overall and in every season in which they were sampled (Table 2 and Figure 4); the sensitivity was higher than all other study sample tissues and also regulatory quadrant-leaf samples.

#### 2.2.2. Statistical Models

In a regression model for the full time series that did not include Q-P tissue (because this was only collected for part of year 2), root and single-peduncle tissues were both significantly, positively associated with a *C*Las-positive result, relative to the reference standard of CDFA-collected old leaf tissue (Table 5). Their estimated odds ratios (OR) of a positive qPCR test were similar: 1.33 (95% CI, 1.07–1.65) and 1.29 (95% CI, 1.03–1.60), respectively. When the same regression model was run for the time period including Q-P tissue, only Q-P was significantly associated with a *C*Las+ result (OR, 1.74; 95% CI, 1.27–2.39). Regarding seasonality, as noted in Section 4.3.2, logistic regression models were run on both datasets to evaluate the interaction of tissue type and season and found no statistically significant effects.

## 3. Discussion

The current management strategy for HLB in California is to control the psyllid population, and detect and remove infected trees as early as possible [12]. Optimizing the sampling protocol will increase the sensitivity of detection, which will lead to the earlier removal of infected trees. In this study, sampling quadrant-peduncle (Q-P) tissue consistently resulted in better *C*Las detection than any other tissue sample tested, both overall and in the season in which they were collected. Single-peduncle and root-tissue samples, although not as accurate as Q-P samples, still had higher detection rates than Q-L (quadrant-leaf) samples. Importantly, a significant number of *C*Las-positive trees would have been misclassified as negative, if using only Q-L sampling or carrying out sampling guided by symptoms.

Quadrant sampling is widely used for increasing the likelihood of detecting pathogens in perennial tree crops. However, using Q-L tissue samples as a standard method for HLB surveillance may result in the failure to detect a significant percentage (30% false-negative rate observed) of *C*Las-positive trees that could be detected by testing other tissues. Based on the presented findings, the best alternative would be Q-P sampling, which had a significantly higher detection sensitivity than Q-L sampling (84% vs. 76%, respectively), as well as an observed positivity rate that was over twice as high as Q-L samples (55% vs. 25%, respectively). Comparing quadrant sampling from different tissues in this way underscores the fact that the high detection accuracy of Q-P tissue was not artificially inflated by having four samples instead of only one. Although Q-P samples were not collected in winter, the observation that single-peduncle samples did not show a large decrease in detection sensitivity in winter (relative to other tissues) suggests that Q-P samples would likewise be consistent.

Peduncle and root-tissue samples had the highest counts of qPCR-positive results of the tissues that were collected throughout the two-year study period. Previous work in Florida and Texas has shown that *C*Las is unevenly distributed in an infected tree [13,14,15,16,17]. Correspondingly, studies have shown that tissues other than the leaf petiole, such as the peduncles of fruit, have higher titers of *C*Las [13,14], and *C*Las is detected earlier, more consistently, and more often in root samples than in leaf samples [17]. In the present study, single-peduncle and root-tissue samples were observed to have a similar probability of detecting *C*Las in a tree that was known to be infected (75% and 76%, respectively). However, from an operational standpoint, the collection of a peduncle sample is easier and less time-consuming than a root sample, especially from residential properties.

Symptoms are often used to guide operational choices about which trees to sample. This strategy may result in a high false-negative rate under California field-sampling conditions. If the presence of symptoms was the criterion for testing trees, 30% of the *C*Las-positive trees would not have been tested. Irey et al. [18] concluded that the timing of sampling is critical in Florida because heightened symptom expression from July through January coincides with an increase in HLB detections. However, owing to the cryptic nature of HLB, there is inherently a broad and variable lag period between *C*Las infection and symptom expression in foliage [19]. Sampling protocols in California that use symptom presence as a requirement for sampling may need to be reevaluated in light of these findings.

In terms of source tissue (old leaf) vs. sink tissue (young leaf, fruit peduncle, feeder root), sink tissues had a higher number of positive results. However, young leaves appeared to be somewhat more consistent with mature leaf source tissues than with the other sink tissues. The data presented here partially support the pressure flow hypothesis that phloem-limited plant pathogens move in a source-to-sink fashion, along with photoassimilates [20,21,22]. The pressure flow hypothesis, or Münch mechanism, states that sugars made in the leaves via photosynthesis move passively down a concentration gradient or are actively transported in some cases [23,24]. Several studies involving citrus pathogens have reported similar observations. Regarding the fruit peduncle as a sink tissue, Bar-Joseph et al. [25] found a higher content of CTV in peduncle bark than in the bark of branches of the same age. Moreover, fruit tissues such as the columella and receptacle are routinely used by the Citrus Pest Detection Program (CPDP) in central California for the reliable detection of *Spiroplasma citri* (another phloem-limited bacterial pathogen) [26]. Furthermore, a relatively high titer of *C*Las has been found in the fruit peduncle [13,14]. Regarding the feeder root as a sink tissue, in one study, the detection of *C*Las occurred earlier, more consistently, and more often in root samples than in leaf samples [17]. Regarding the young leaf as a sink tissue, new flush tissue has been a reliable tissue for testing *Citrus tristeza virus* (CTV), and the CPDP has been utilizing flush tissue for large-scale field surveys over the last three decades [27]. However, the contrasting results observed in this study on *C*Las detection in young leaf tissue, compared with other sink tissues, are possibly due to tissue tropism and/or the differences in the interaction of viruses and bacteria with their respective host factors in systemic movement.

One goal of this study was to determine whether different tissue types might be more accurate in certain seasons, which would make it possible to optimize tissue sampling protocols by knowing which was the best tissue to sample in each season. However, it was found that Q-P samples had the highest detection accuracy in the three seasons that they were collected. Single-peduncle and root-tissue samples had consistently higher *C*Las-positive counts and Se than leaf tissue in every season. The single-peduncle and root-tissue samples did switch places in different seasons; for example, roots had a higher Se in spring and single-peduncle tissue had a higher Se in fall. To determine if these observed results were likely to be a random variation or if they might be generalizable outside this study, the observed data were statistically modeled. The results demonstrated that there were no statistically significant relationships between season and tissue type. Thus, based on these findings, there was no evidence that certain sink tissues are relatively more likely to be *C*Las-positive in a certain season.

A major strength of this research is that the study samples were collected and processed side-by-side with regulatory samples. Because the study design, in many ways, mimicked operational practices by piggybacking on regulatory sampling, it is possible to apply the findings more confidently to those operational practices than result from a controlled experimental study. The methodology controlled variations in collection timing, sampling technique, field staff idiosyncrasies, and tissue handling and processing. However, the study design—observational rather than experimental—did limit the questions that could be addressed in the analysis, e.g., it could not be addressed which tissues might be better for detecting pathogens at an early vs. late stage of infection because there was no verifiable way in this study of determining when a tree with a *C*Las-positive sample was infected. It also introduced an unknown amount of random noise into the data, which was accounted for by incorporating analytic methods that support making inferences from the observations.

## 4. Materials and Methods

### 4.1. Plant Material

Plant material was collected in collaboration with the California Department of Food and Agriculture (CDFA) by conjointly sampling during its regulatory activities. CDFA field staff collected the plant samples and the CDFA Plant Pest Diagnostics Center (PPDC) in Sacramento provided lyophilized tissues to the Citrus Pest Detection Program (CPDP) in Tulare for DNA extraction and testing by qPCR. This sampling protocol allowed direct comparison between regulatory samples—i.e., mature leaf samples collected for CDFA disease-control purposes—and study samples, because all samples were collected at the same time, under the same conditions, and from the same trees.

A total of 765 residential citrus-tree-sampling events were conducted from 408 trees between 7 November 2019 and 26 October 2021. Samples were collected in Southern California counties, i.e., Los Angeles, Orange, Riverside, San Bernardino, San Diego, and Ventura. Sites were located within HLB quarantine zones and, secondarily, outside those zones. All trees tested were residential citrus trees chosen by the CDFA as part of its HLB-control activities, including: (a) trees that had previously tested positive and were sampled only one time before being removed; (b) trees that had previously tested inconclusive (with a Ct value over 38); (c) trees from *C*Las-positive ACP sites that were putatively exposed to *C*Las but were qPCR-unconfirmed; (d) remaining adjacent trees on a previous-find property; and (e) trees alerted by the K9 HLB detection dog team (Appendix A). Trees other than the known positives were re-sampled for study samples every season until a tree tested positive and was removed or until the research project was complete.

Four tissue types were collected for the purposes of the study: (1) mature leaves at least one growth period old; (2) young leaves; (3) fruit peduncle bark or stem bark; and (4) feeder roots (Figure 5). For regulatory purposes, the CDFA collected mature leaf samples at every site. It also sometimes collected quadrant-leaf (QL) samples; this was done on a case-by-case basis by assessing the exposure risk and cooperation from the property owner. Quadrant-peduncle (Q-P) sampling was not a part of the original research proposal. After reviewing the results of year 1, the study protocol was modified to include Q-P sampling for a portion of year 2 of the project, between April and October 2021. When Q-L and/or Q-P samples were taken, this was carried out by collecting four independent samples per tree, from the north, south, east, and west sides of the tree. See Box 1 for a summary of each sample type.

Box 1Summary of plant tissue samples.Quadrant mature leaf sample (Q-L): Four samples per tree, consisting of 16 mature leaves per sample.Old leaf: Single sample (16 old leaves per tree).Young leaf: Single sample (16 young leaves per tree).Peduncle: Single sample (8 peduncles per tree).Roots: Single sample (feeder roots collected around the tree canopy as one sample).Quadrant peduncle (Q-P): Four samples per tree, consisting of 8 peduncles per sample.

### 4.2. CLas Testing

Approximately 150 mg of tissue was used for DNA extraction. The leaf petiole, fruit peduncle, and stem bark tissue were chopped and lyophilized overnight. Similarly, feeder root samples were washed and dried, and approximately 150 mg of tissue was chopped and lyophilized overnight. At the PPDC lab in Sacramento, DNA was extracted using the USDA regulatory work instructions (USDA WI-B-T-1-53) (Park et al., 2018). At the CPDP lab in Tulare, DNA was extracted from the lyophilized samples using NucleoMag^®®^ Plant DNA Isolation kit (Macherey-Nagel Inc., Düren, Germany) in the automated DNA extraction instrument (KingFisher^TM^ Flex by ThermoFisher Inc., Waltham, MA, USA) The mature leaves and quadrant-leaf samples were tested using the USDA validated ribonucleotide reductase (RNR) qPCR assay (USDA WI-B-T-1-55) (Appendix A) [11]. A total of 4238 samples were processed and tested by the CPDP in Tulare. In addition, 920 quadrant-leaf samples were processed and tested simultaneously by the PPDC lab in Sacramento. Then, qPCR Ct values were recorded and used for the analyses.

### 4.3. Data Analysis

#### 4.3.1. Detection Sensitivity of Tissue Types and of Symptoms

To evaluate how accurately each tissue sample type identified infected trees, known-positive trees were used as a reference. The known positives were trees that had a qPCR-positive sample previously collected and tested by the CDFA. This allowed calculation of the sensitivity (Se), i.e., the likelihood of detecting the pathogen in a tree that was known to be infected, for each tissue type.

The Se was also calculated for the presence of symptoms, to evaluate the detection sensitivity of using symptoms to guide sampling choices. A record of the presence or absence of symptoms was available for 601 sampling events in the dataset. Rather than evaluating symptom data by comparing them with previous known positives, they were compared with the qPCR results of all tissue samples collected at the same time that symptoms were observed to be present/absent. This was done to control the possibility of changes in the symptomatic status between prior qPCR test results and the time symptoms were noted. If any of the tissues from the sampling event were qPCR-positive, the tree was coded as infected, for the purposes of evaluating the likelihood that symptomology would accurately reflect infection status.

To quantify the detection accuracy of the Q-L samples collected for regulatory purposes, the qPCR results from the Q-L samples were compared against the positive/negative qPCR results from all study tissues combined. If any of the study tissues were found to be *C*Las-positive, the true disease status of the tree was assumed to be infected. The Q-L sample results were then compared against these to calculate the sensitivity.

Where a reference quadrant-leaf sample from the CDFA was available, the positivity rate of each study tissue type (excluding Q-P) was calculated when there were different numbers of positive quadrants; for example, the positivity rates of study samples from trees where only one quadrant was positive, the positivity rates when two quadrants were positive, etc. The purpose of this was to use the number of positive quadrants as a proxy index of how widely distributed *C*Las was throughout the tree. Thus, the simplifying assumption was made that a tree with four positive quadrants had a more advanced, extensive infection than a tree with only one positive quadrant. This was used to evaluate the relative utility of different types of sample tissues for detecting *C*Las at various stages of infection.

Some analyses that included Q-P tissue, including seasonal effects analyses and statistical models below, had to be conducted separately because this sample tissue type was only collected between April and October 2021.

#### 4.3.2. Seasonal Effects

The data were disaggregated by season and the Se was recalculated to determine whether certain tissue types were more or less accurate in different seasons. Seasons were defined as four calendar-month periods, beginning with March, April, and May as the spring season. The relationship between season and detection accuracy for each study tissue type was further assessed using statistical modeling (see Section 2.2.2 for additional information on modeling methods).

#### 4.3.3. Statistical Models

Statistical modeling was used to estimate the relative likelihood of *C*Las being detected in each of the study sample tissues during each sampling event. This was estimated with general linear models with the assumption of a binomial error distribution and logit link function.

First, the association between the tissue sample types and their relative likelihood of a positive result was modeled to evaluate statistically the generalizability of the observed study data. CDFA-collected samples of old leaf tissue were used as a comparison against which to evaluate the relative outcomes of the study sample tissues. Next, seasonal effects were assessed. The season and tissue type were modeled as interactive effects, which assessed whether certain tissue types were more or less likely to be qPCR-positive in different seasons (e.g., if roots were more likely than leaves to be positive in winter).

All data analyses and statistical models were conducted using R, version 4.1.1 [28]. Statistical significance was assessed at α = 0.05.

## 5. Conclusions

Quadrant-peduncle (Q-P) tissue sampling consistently resulted in better *C*Las detection than any other tissue type.Q-P samples had a 30% higher qPCR positivity rate than quadrant-leaf (Q-L) samples.No significant seasonal patterns were observed.Roots and single peduncles had similar detection rates to each other; both were higher than single leaves or Q-L samples.If symptoms were used to guide sampling, 30% of infected trees would have been missed.Q-P tissue sampling is the optimal choice for improved *C*Las detection.

Manjunath et al. [29] recorded the spread of *C*Las in an area by detecting the pathogen in its insect vector, *D. citri*, one to several years before the development of HLB symptoms in plants. Based on these findings and the current study, to improve *C*Las detection, an ideal sampling method (Figure 6) would involve:Prioritizing ACP sampling for early *C*Las detection.If symptoms are found → sample the symptomatic tissue and, importantly, also sample nearby peduncle tissue.If symptoms are not present, but there is an indication of possible *C*Las exposure—e.g., nearby positive tree, nearby ACP finds, etc. → collect a quadrant-peduncle tissue sample.If all qPCR results are negative but follow-up sampling is required → sample root tissue for the follow-up. Although roots are not the most diagnostically sensitive tissue in general, there is some evidence that they may be preferable in cases where all aboveground tissue has tested negative.

## Figures and Tables

**Figure 1 plants-12-03364-f001:**
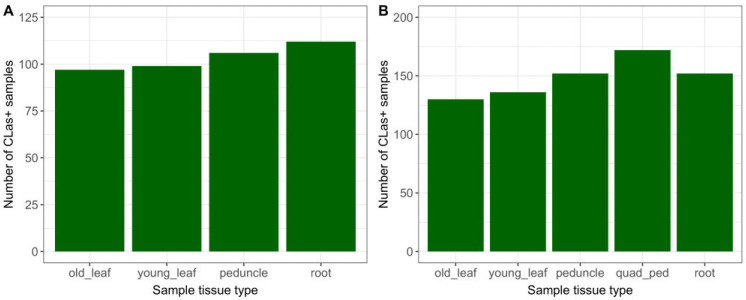
*C*Las+ detections by sample tissue type, for (**A**) 451 sampling events, November 2019–April 2021; and (**B**) 314 sampling events, April–October 2021 (including Q-P).

**Figure 2 plants-12-03364-f002:**
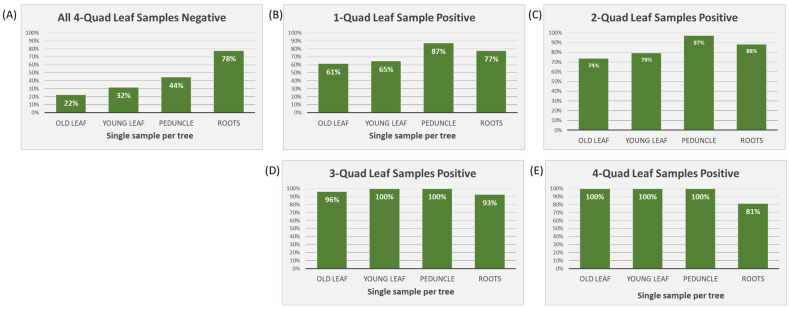
Percentage of positive samples (of young leaves, old leaves, fruit peduncles, and feeder roots) in cases when between zero and all four samples included in the quadrant leaf sample tested positive (*n =* 642). (**A**) All four quadrant leaf samples tested negative (*n =* 479); (**B**) only one quadrant leaf sample tested positive (*n =* 36); (**C**) two quadrant leaf samples tested positive (*n =* 36); (**D**) three quadrant leaf samples tested positive (*n =* 28); and (**E**) all four quadrant leaf samples tested positive (*n =* 63).

**Figure 3 plants-12-03364-f003:**
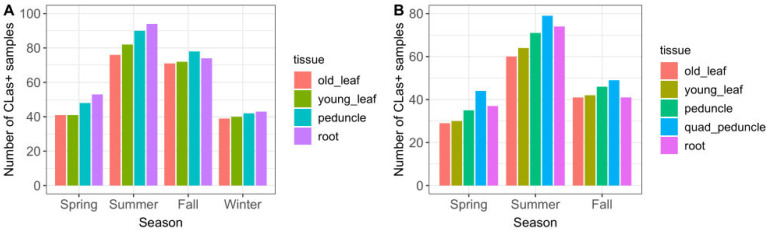
*C*Las-positive detections by season and sample tissue type, for (**A**) 765 sampling events, November 2019–October 2021 (full time series excluding Q-P); and (**B**) 314 sampling events, April–October 2021 (subset of time series including Q-P, which was not collected during any winter months). The full two-year time series is shown in the first figure so that there are an equal number of seasons (two years of each season); showing only November 2019–April 2021 results in there being two years of winter season data, but only one year of the other seasons, which would inaccurately skew the figure with an excess count in the winter season.

**Figure 4 plants-12-03364-f004:**
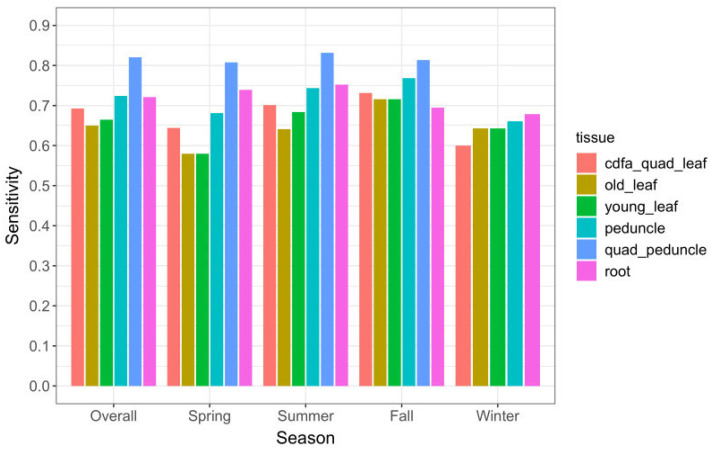
Sensitivity (Se) of tissue sample types overall and by season for 765 sampling events, November 2019–October 2021. Q-P samples do not appear in the winter category because that tissue type was not collected during any winter months.

**Figure 5 plants-12-03364-f005:**
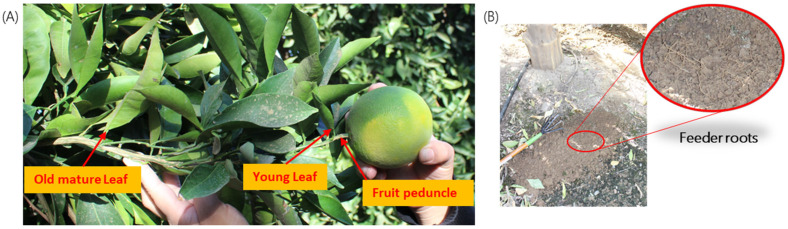
Four tissue types: (**A**) old and young leaves, fruit peduncle/bark, and (**B**) feeder roots (shown in the zoomed picture in the subfigure) were used to test the relative titer of ‘*Candidatus Liberibacter* asiaticus’ in all four seasons.

**Figure 6 plants-12-03364-f006:**
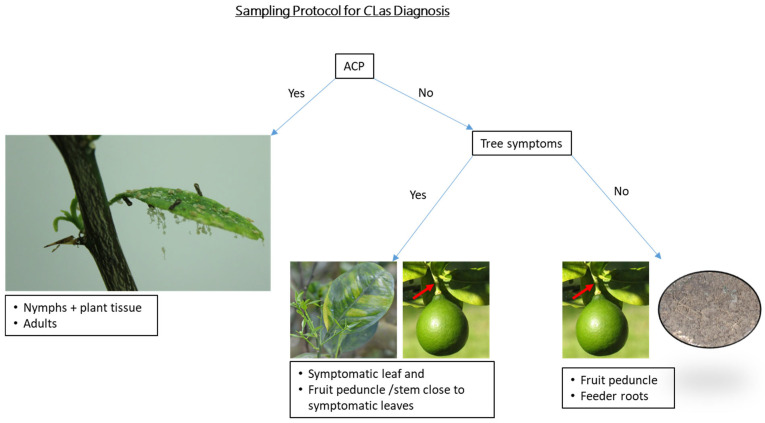
Flow chart illustrating an ideal sampling method for improved *Candidatus Liberibacter* asiaticus detection.

**Table 1 plants-12-03364-t001:** Summary of qPCR results of ‘*Candidatus* Liberibacter asiaticus’ (*C*Las) testing for tissue samples collected from November 2019 to October 2021. See Methods for full description of samples.

Type	N	% Pos (CLas+/Total)	Mean Ct Value
Trees	408	89.7% (366/408)	NA
Sampling events	765	48.2% (369/765)	NA
Tissue Samples	N	% Pos (CLas+/Total)	Mean Ct value
CDFA original sample ^a^	765	Trees: 82.6% (337/408) Samples: 45.2% (346/765)	26.5
CDFA quadrant leaf ^b^	642	25.2% (162/642)	25.7
Old leaf	765	29.7% (227/765)	24.9
Young leaf	765	30.7% (235/765)	25.2
Single peduncle	765	33.7% (258/765)	24.8
Quadrant peduncle ^c^	314	54.8% (172/314)	26.0
Root	765	34.5% (264/765)	26.7

^a^ Out-of-study sample of old leaf, collected at an earlier time point. ^b^ Inconsistently collected throughout the study period. Quadrant leaf samples were not collected from every tree that was used for sampling four tissue types (old leaf, young leaf, peduncle, and root). ^c^ Consistently collected from April to October 2021. Quadrant-peduncle samples were collected from every tree that was used for sampling four tissue types (old leaf, young leaf, peduncle, and root).

**Table 2 plants-12-03364-t002:** *C*Las detection sensitivity (Se) of sample tissues, using known-positive trees as the reference standard. Se is calculated here by dividing the number of positive samples for each tissue type by the total number of known-positive trees, to give an observed likelihood of that sample tissue type accurately detecting the infection.

Tissue Sample	Overall	Spring	Summer	Fall	Winter
Study samples					
Old leaf	65.6% (227/346)	58.6% (41/70)	63.3% (76/120)	71.7% (71/99)	68.4% (39/57)
Young leaf	67.9% (235/346)	58.6% (41/70)	68.3% (82/120)	72.7% (72/99)	70.2% (40/57)
Single peduncle	74.6% (258/346)	68.6% (48/70)	75.0% (90/120)	78.8% (78/99)	73.7% (42/57)
Quadrant peduncle	83.5% (172/206)	84.6% (44/52)	83.2% (79/95)	83.1% (49/59)	NA
Root	76.3% (264/346)	75.7% (53/70)	78.3% (94/120)	74.7% (74/99)	75.4% (43/57)
CDFA samples collected at the same time point as study samples					
CDFA Quadrant leaf	72.3% (162/224)	64.4% (38/59)	69.7% (62/89)	71.4% (50/70)	66.7% (4/6)

**Table 3 plants-12-03364-t003:** Cross-tabulation of symptoms and of positive/negative qPCR results from study sample tissues. The plant sample results were categorized as positive if any of the study and/or regulatory sample tissues collected during the sampling event were positive. The purpose of this was to identify trees known to be positive at the time when symptomology was recorded, in order to quantify symptomology Se, i.e., the likelihood of symptoms accurately identifying an infected tree.

Symptoms	Plant Sample Tissue		Total
	Pos	Neg	
Yes	182	146	328
No	78	195	273
Total	260	341	601

**Table 4 plants-12-03364-t004:** Cross-tabulation of positive/negative qPCR results from quadrant-leaf samples (Q-L) and from study sample tissues. The study sample results were categorized as positive if any of the tissues were positive. The purpose of this was to identify trees known to be positive at the time of Q-L sample collection, in order to quantify Q-L tissue Se, i.e., the likelihood of Q-L tissue positively identifying an infected tree.

Q-L	Study Sample Tissue		Total
	Pos	Neg	
Pos	162	1	163
Neg	69	410	479
Total	231	411	642

**Table 5 plants-12-03364-t005:** Parameter estimates for sample tissue type logistic regression models. Model 1 uses time series from November 2019–October 2021; Model 2 uses time series from April–October 2021, when Q-P sample tissue was collected.

Parameter	OR (95% CI)	*p*
Model 1: qPCR result (pos/neg)~tissue sample type (excl. quadrant peduncle)		
CDFA-collected old leaf	1.00 (reference)	NA
Old leaf (study sample)	1.07 (0.85–1.33)	0.57
Young leaf	1.12 (0.90–1.40)	0.31
Peduncle	1.29 (1.03–1.60)	0.020 *
Root	1.33 (1.07–1.65)	0.0097 **
Model 2: qPCR result (pos/neg)~tissue sample type (incl. quadrant peduncle)		
CDFA-collected old leaf	1.00 (reference)	NA
Old leaf (study sample)	1.01 (0.74–1.39)	0.94
Young leaf	1.10 (0.80–1.50)	0.57
Peduncle	1.35 (0.98–1.85)	0.065
Quadrant peduncle	1.74 (1.27–2.39)	0.00062 ***
Root	1.35 (0.98–1.85)	0.065

Abbreviations: CI, confidence interval; OR, odds ratio. *, **, and *** indicate statistical significance at *p* < 0.05, *p* < 0.01, and *p* < 0.001, respectively.

## Data Availability

The data presented in this study are available on request from the corresponding author. The data are not publicly available due to privacy restrictions.

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
