# Peer review of "Alternative Tissue Sampling for Improved Detection of Candidatus Liberibacter asiaticus"

_plants, 2023, doi:10.3390/plants12193364_

Round 1

Reviewer 1 Report

Alternative Tissue Sampling for Improved Detection of Candi-
datus Liberibacter asiaticus

This study is well designed, well executed and generally well written. The methods and analysis used justify the results which lead to important recommendations for early detection of CLas in citrus trees, which is crucial to manage and limit the spread of HLB in California. I recommend 'acceptance with minor changes' that include answering the following questions and/or adopting the following suggestions to improve the  accuracy and readability of this ms.:
Line 333: "3) fruit peduncle bark or stem bark". Why is "stem bark" mentioned only here in the methods section but not in the results or discussion? Was stem bark used as an alternative to peduncle sometimes, why and when?!
Were the results of stem bark samples pooled with the peduncle samples, if so what was the ratio of each? Was there any significant differences in detection sensitivity between these 2 categories?

Other suggestions:
L. 23-24. Delete "in every season" because Q-P was not tested in winter.
L. 46. Add some reference(s).
L. 79. Delete "in every season".
L. 107. Replace "in each season" with "in the three seasons tested".
L. 287. Replace "in every season" with "in the three seasons".

Fig. 3: The color used for each tissue type should preferably be consistent in Figs. 3a and 3b.

Table 5. Please clarify: Does model 2 include only the time period during which Q-P was tested or the whole study periodof 2 years?  

Author Response

REVIEWER 1

Comments and Suggestions for Authors

Alternative Tissue Sampling for Improved Detection of Candidatus Liberibacter asiaticus

This study is well designed, well executed and generally well written. The methods and analysis used justify the results which lead to important recommendations for early detection of CLas in citrus trees, which is crucial to manage and limit the spread of HLB in California. I recommend 'acceptance with minor changes' that include answering the following questions and/or adopting the following suggestions to improve the  accuracy and readability of this ms.:

Line 333: "3) fruit peduncle bark or stem bark". Why is "stem bark" mentioned only here in the methods section but not in the results or discussion? Was stem bark used as an alternative to peduncle sometimes, why and when?!
Were the results of stem bark samples pooled with the peduncle samples, if so what was the ratio of each? Was there any significant differences in detection sensitivity between these 2 categories?

Quadrant-bark tissue was collected in place of quadrant-peduncle tissue for two trees (out of 765 trees sampled). Both trees (#742 & #744) were negative for CLas with all tissue types. It was in the protocol to collect bark tissue of new growth in cases where the trees are small, and/or fruit is not present, or the homeowner doesn’t allow harvesting fruit to collect peduncle tissues.

Other suggestions:
L. 23-24. Delete "in every season" because Q-P was not tested in winter.

Done. Deleted and highlighted in the provided manuscript.

  1. 46. Add some reference(s).

Reference added and highlighted in the provided manuscript.

  1. 79. Delete "in every season".

Done. Deleted and highlighted in the provided manuscript.

  1. 107. Replace "in each season" with "in the three seasons tested".

Done. Deleted and highlighted in the provided manuscript.

  1. 287. Replace "in every season" with "in the three seasons".

Done. Deleted and highlighted in the provided manuscript.

Fig. 3: The color used for each tissue type should preferably be consistent in Figs. 3a and 3b.

In software R, the program automatically chooses colors and cannot change individual bar color. Because the two graphs have different tissue types, the program provides different color selections but retains similar color tones.

Table 5. Please clarify: Does model 2 include only the time period during which Q-P was tested or the whole study periodof 2 years?  

Clarification is provided in the table description and highlighted in the provided manuscript.

Reviewer 2 Report

The authors developed a method to improve the detection rate of Candidatus Liberibacter asiaticus (CLas ) causing Huanglongbing (HLB). There are some comments to improve this manuscript:

1. In Figure 2, the ordinate axis should be added to each chart.

2. In Pages 6, ‘but with small variations between them across sea-sons (Table 2 and Figure 5). ’, Figure 5 is right? Please check it carefully and confirm it.

3. Some information about the results should be added to the Conclusions section.

4. In the References section, some information is missing, e.g., references 1 and 2. Each reference should be carefully checked, and missing information should be added.

5. The format of the references in the References section should be checked.

Author Response

REVIEWER 2

Comments and Suggestions for Authors

The authors developed a method to improve the detection rate of Candidatus Liberibacter asiaticus (CLas) causing Huanglongbing (HLB). There are some comments to improve this manuscript:

  1. In Figure 2, the ordinate axis should be added to each chart.

Added.

  1. In Pages 6, ‘but with small variations between them across sea-sons (Table 2 and Figure 5). ’, Figure 5 is right? Please check it carefully and confirm it.

It is Figure 4. Corrected and highlighted in the provided manuscript.

  1. Some information about the results should be added to the Conclusions section.

Results are added to the conclusion section with bullet points.

  1. In the References section, some information is missing, e.g., references 1 and 2. Each reference should be carefully checked, and missing information should be added.

These two references were published as a Special Topic in the Journal of Citrus Pathology. Added this to the reference.

  1. The format of the references in the References section should be checked.

Done.

Reviewer 3 Report

The paper describes about the early detection of Huanglongbing (HLB) in California. Among the different tissues, old and young leaves, peduncle bark of fruit, and feeder roots, Quadrant-peduncle (Q-P) tissue sampling consistently resulted in better CLas detection.  Q-P samples had a 30% higher qPCR positivity rate than quadrant-leaf (Q-L) samples. The paper is written well.  There are some questions which arise during the review can be carried out.

Detail about the qPCR (USDA validated Ribonucleotide Reductase (RNR) qPCR assay) and the primers can be appended eventhough it is published earlier which makes the reader to follow. (Zhang et al ., 2016)

What is the time of collection of samples can be highlighted. What is the age of the tree.? 

The symptoms of CLas at different stages can be given?

What is the probable time of infection and sourse of infection can be discussed to reduce the spread

Whether the vectors have been analysed for the presence of CLas by qPCR

What about the prevelance of vector during off-season or unfavorable conditions. 

Why in Summer the CLas+ is more irrespective of sampling type . Is it due to prevelance of vector 

Author Response

REVIEWER 3

Comments and Suggestions for Authors

The paper describes about the early detection of Huanglongbing (HLB) in California. Among the different tissues, old and young leaves, peduncle bark of fruit, and feeder roots, Quadrant-peduncle (Q-P) tissue sampling consistently resulted in better CLas detection.  Q-P samples had a 30% higher qPCR positivity rate than quadrant-leaf (Q-L) samples. The paper is written well.  There are some questions which arise during the review can be carried out.

Detail about the qPCR (USDA validated Ribonucleotide Reductase (RNR) qPCR assay) and the primers can be appended eventhough it is published earlier which makes the reader to follow. (Zhang et al ., 2016)

Primer/probe sequences are added in supplementary table 2.

What is the time of collection of samples can be highlighted. What is the age of the tree.?

We are unsure about the reviewer’s question regarding the “time of collection of samples”? Is this question directed at the ‘time of the year’? If so, the seasonality was considered for the analysis.

The age of the trees varied from very young (less than 5 ft) to medium (6-10 ft) to large (more than 10 ft) tall trees. In some instances, the tree age/height data was not captured. We didn’t include this data in our analysis.

The symptoms of CLas at different stages can be given?

Since this study was observational rather than a controlled study, it was not possible to capture the symptoms of CLas at different stages of infection.

What is the probable time of infection and sourse of infection can be discussed to reduce the spread.

ACP has been established in southern California counties; hence, trapping has been abandoned. No consistent data is available to connect ACP find or prevalence data to the probable time and source of infection.

Whether the vectors have been analysed for the presence of CLas by qPCR.

ACP samples were not analyzed as part of this study.

What about the prevelance of vector during off-season or unfavorable conditions.

It is a good question, and it needs to be analyzed to see the impact of the prevalence of CLas-positive ACP during off-seasons. However, a detailed analysis of host characteristics and vector dynamics is beyond the scope of the present study. 

Why in Summer the CLas+ is more irrespective of sampling type . Is it due to prevelance of vector.

Like the previous question, it is a good hypothesis, and it warrants further analysis to connect the dots in this host-vector-pathogen dynamics in the epidemiology of the disease at large.